# Uncertainty-Aware Diagnostics for Physics-Informed Machine Learning

**Mara Daniels**
Department of Mathematics
Massachusetts Institute of Technology
maradan@mit.edu

**Liam Hodgkinson**
School of Mathematics and Statistics
University of Melbourne
lhodgkinson@unimelb.edu.au

**Michael W. Mahoney**
ICSI, LBNL, Department of Statistics
University of California at Berkeley
mmahoney@stat.berkeley.edu

## Abstract

Physics-informed machine learning (PIML) integrates prior physical information, often in the form of differential equation constraints, into the process of fitting ML models to physical data. Popular PIML approaches, including neural operators, physics-informed neural networks, and neural ordinary differential equations, are typically fit to objectives that simultaneously include both data and physical constraints. However, the multi-objective nature of this approach creates ambiguity in the measurement of model quality. This is related to a poor understanding of epistemic uncertainty, and it can lead to surprising failure modes, even when existing metrics suggest strong fits. Working within a Gaussian process regression framework, we introduce the Physics-Informed Log Evidence (PILE) score. Bypassing the ambiguities of test losses, the PILE score is a single, uncertainty-aware metric that provides a selection principle for hyperparameters of a physics-informed model. We show that PILE minimization yields excellent choices for a wide variety of model parameters, including kernel bandwidth, least squares regularization weights, and even kernel function selection. We also show that, prior to data acquisition, a special "data-free" case of the PILE score identifies a-priori kernel choices that are "well adapted" to a given PDE. Beyond the kernel setting, we anticipate that the PILE score can be extended to PIML at large, and we outline approaches to do so.

## 1 Introduction

A great challenge in machine learning (ML) in general and Scientific ML (SciML) in particular involves the development of models that can combine in principled ways data-driven information (as is common in ML) and domain-driven information (as is common in physical and other sciences). Strategies that attempt to achieve this are grouped under the umbrella of *physics-informed machine learning* (PIML). These approaches consist of general purpose tools for scientific computation that also enjoy the scalability and flexibility of high-dimensional ML. PIML methods include *Physics-Informed Neural Networks* (PINNs) (Raissi et al., 2019) (which led to a large body of empirical (Krishnapriyan et al., 2021; Karniadakis et al., 2021; Sirignano & Spiliopoulos, 2018; Sahli Costabal et al., 2020; Jin et al., 2021; Geneva & Zabaras, 2020; Xu & Darve, 2020) and theoretical (Minakowski & Richter, 2023; Lu et al., 2021; Doumèche et al., 2024b) analyses), *Neural Ordinary Differential Equations* (Neural ODEs) (Chen et al., 2018; Krishnapriyan et al., 2023), *Neural Operators* (Kovachki et al., 2023), and *Neural Discrete Equilibrium* (NeurDE) (Benitez et al., 2025). Unfortunately, PINNs and related methods are notoriously difficult to train (Krishnapriyan et al., 2021); they lack robust *a posteriori* error estimates that are typically available for classical numerical partial differential equation (PDE) solvers; and they lack a strong grounding in statistical theory. These issues are exacerbated by the multi-objective nature of PINNs, as well as other PIML methods that incorporate domain knowledge as a soft regularization.

It is (easy and thus) popular to view physical constraints in terms of regularizers, in a manner analogous to ridge regression (Karniadakis et al., 2021). However, practical implementations involve a delicate trade-off between errors to noisy observations and adherence to the imposed (or assumed) physical equations. Indeed, one interpretation of the "failure modes" results of Krishnapriyan et al. (2021) is that while reducing physical error is of interest, this strategy becomes nuanced (and error-prone) when the model is unable to satisfy the physical constraints perfectly, or when the constraints are misspecified. In these cases, without sufficient validation data (which is common in scientific settings), it becomes challenging to determine whether a model is a suitable fit. These challenges are not limited to neural network models—they are common to many other methods that aim to combine data-driven ML models with domain-driven physical models. **It is critical to understand how to quantify the quality of PIML models**, in a manner analogous to how one quantifies model quality in statistical learning theory.

In this paper, we provide a first step toward solving this PIML model selection problem, addressing the problem under the Physics-Informed Kernel Learning (PIKL) framework (Pförtner et al., 2024). PIKL considers a Gaussian process (GP) model for solving linear PDEs under known conditions, and it offers a powerful, uncertainty-aware approach to PIML. One advantage of the GP framework is that it offers a structured, probabilistic approach that allows for rigorous uncertainty quantification (UQ) (which is often missing in other physics-informed models[1]). Another advantage of GPs lies in their ability to seamlessly incorporate multiple forms of data acquisition, including noisy pointwise observations of the solution and derivative data derived from the governing PDE. This flexibility enables a principled integration of prior knowledge about the system, while maintaining a Bayesian framework for uncertainty estimation.

Our main contributions are as follows:

(I) We introduce a **model selection criterion** called the *Physics-Informed Log Evidence* (PILE) (Section 3). The PILE criterion provides a theoretically-grounded way to assess the suitability of different kernel choices for PIML tasks; and it can be used, e.g., to optimize hyperparameters of the GP model, including the kernel function, its bandwidth, and regularization parameters.

(II) We provide an **empirical evaluation** demonstrating that the PILE criterion is a reliable indicator of model performance (Section 5). Models optimized using PILE exhibit strong predictive accuracy and adherence to physical constraints. By studying the challenging wave-equation setting introduced in Krishnapriyan et al. (2021), we show that the PILE score can not only diagnose model misspecification, but it can be used to identify the 'best' kernel function for the problem at hand, leading to vastly improved performance.

Overall, **we claim that free energy metrics provide the solution to the multi-objective bottleneck in PIML**, establishing a single number that can be optimised to ensure *both* a strong fit to existing data *and* adherence to a governing differential equation. Diagnostics using free energy can be conducted both *a priori* using our Fredholm determinant metric, at the stage of architecture selection before fitting to data, and *a posteriori* using our PILE score, after the model has been fitted to data. Our case studies demonstrate how the use of these tools can bypass well-known pitfalls in PIML, highlighting scenarios where a model choice will lead to an undesirable fit.

## 2 BACKGROUND AND RELATED WORK

### 2.1 LINEAR PARTIAL DIFFERENTIAL EQUATIONS

An $s$-th order linear differential operator $\mathcal{D} : C^s(\Omega) \to C^0(\Omega)$ for integer $s \geq 1$ has the form

$$\mathcal{D}f(x) = \sum_{\|\alpha\|_1 \leq s} c_\alpha(x) \frac{\partial^{\alpha_1}}{\partial x_1^{\alpha_1}} \cdots \frac{\partial^{\alpha_d}}{\partial x_d^{\alpha_d}} f(x),$$

where $\Omega$ is an open, bounded subset of $\mathbb{R}^d$ with $C^1$ boundary, where $\alpha \in \mathbb{Z}_{\geq 0}^d$ is a multi-index, and where $\{c_\alpha : \|\alpha\|_1 \leq s\}$ are $C^0(\Omega)$ coefficient functions. Given $g \in C^0(\Omega)$ and $h \in C^0(\partial\Omega)$, we say

---

[1]Some exceptions exist, including Neural Processes (Garnelo et al., 2018; Kim et al., 2019), but they will prove to be disadvantageous in our framework due to intractable marginal likelihoods.

that $f$ solves the *Dirichlet boundary value problem* (BVP) if

$$\mathcal{D}f(x) = g(x) \text{ for } x \in \Omega, \qquad f(x) = h(x) \text{ for } x \in \partial\Omega. \tag{1}$$

Under our assumptions, $f$ solves the Dirichlet BVP if and only if it minimizes the energy functional

$$\mathcal{E}(f) := \|\mathcal{D}f - g\|_{L^2(\Omega)}^2 + \|f - h\|_{L^2(\partial\Omega)}^2. \tag{2}$$

This variational problem is a key ingredient in the formulation of equation 1 as a ML problem. Note, however, that we can extend our formulation far beyond the Dirichlet setting to encompass a wide range of mixed boundary conditions. To simplify matters, observe that the formulation is no less general if $g = 0$, as we can instead take the differential operator $\mathcal{D} - g\mathsf{Id}$. Let $\mathcal{D}$ be as before, but now let $\mathcal{B}_i : C^s(\Omega) \to C^0(\Gamma_i)$, $i = 1 \ldots p$ be a family of operators, where each $\Gamma_i \subset \partial\Omega$. We can now consider the general mixed boundary condition

$$\mathcal{D}f(x) = 0 \text{ for } x \in \Omega, \qquad (\mathcal{B}_i f)(x) = 0 \text{ for } x \in \Gamma_i, \ i = 1 \ldots p$$

This formulation can encode the following boundary conditions:

- **Dirichlet:** $p = 1$, $\Gamma_1 = \partial\Omega$, and $\mathcal{B}_1 f = (f - h)|_{\partial\Omega}$ is the restriction of f to $\partial\Omega$.
- **Neumann:** $p = 1$, $\Gamma_1 = \partial\Omega$, and $\mathcal{B}_1 f = (\nu \cdot \nabla f - h)|_{\partial\Omega}$, where $\nu$ is the unit normal to $\partial\Omega$.
- **Robin:** $p = 1$, $\Gamma_1 = \partial\Omega$, and $\mathcal{B}_1 f = (af + b\nu \cdot \nabla f - h)|_{\partial\Omega}$ for $a, b \in \mathbb{R}$.
- **Cauchy:** $p = 2$, $\Gamma_1 = \Gamma_2 = \partial\Omega$, $\mathcal{B}_1 f = (f - h_1)|_{\Gamma_1}$, and $\mathcal{B}_2 f = (\nu \cdot \nabla f - h_2)|_{\Gamma_2}$.

Now any solution $f$ is a minimizer of the energy functional

$$\mathcal{E}(f) := \|\mathcal{D}f\|_{L^2(\Omega)}^2 + \sum_{i=1}^{p} \|\mathcal{B}_i f\|_{L^2(\Gamma_i)}^2. \tag{3}$$

Letting $\mathcal{A}f(x) = (\mathcal{D}f(x)\mathbf{1}_\Omega, \mathcal{B}_1 f(x)\mathbf{1}_{\Gamma_1}, \ldots, \mathcal{B}_p f(x)\mathbf{1}_{\Gamma_p}) \in \mathbb{R}^{p+1}$, we can define a measure $\mu$ that is Lebesgue on $\Omega$ and Hausdorff on each $\Gamma_i$ so that $\mathcal{E}(f) = \|\mathcal{A}f\|_{L^2(\bar{\Omega}, \mu, \mathbb{R}^{p+1})}^2$. This notation will become convenient later. While integral constraints can also be incorporated naturally within this setup (Hansen et al., 2023), but here we restrict attention to differential operator constraints to avoid complicating our analysis.

## 2.2 GAUSSIAN PROCESS REGRESSION

A *Gaussian process (GP)* $f$ on $\Omega \subseteq \mathbb{R}^d$, denoted $f \sim \mathcal{GP}(m, k)$, is a stochastic process where for some *mean function* $m : \Omega \to \mathbb{R}$ and a *kernel function* $k : \Omega \times \Omega \to [0, \infty)$, any projection onto finitely many points $X = \{x_i\}_{i=1}^n \subseteq \Omega$ is multivariate Gaussian:

$$(f(x_1), \ldots, f(x_n)) \sim \mathcal{N}((m(x_i))_{i=1}^n, (k(x_i, x_j))_{i,j=1}^n).$$

As covariance matrices are necessarily symmetric positive semi-definite, we require that the *Gram matrix* $(k(x_i, x_j))_{i,j=1}^n$ is a positive semi-definite matrix, for any $\{x_i\}_{i=1}^n$. (Any function $k$ with this property is said to be positive semi-definite.) We further require $k$ to be continuous on $\Omega \times \Omega$ and to have $\int_\Omega k(x, x)dx < \infty$.

GP regression is a framework which provides a Bayesian perspective on kernel regression, along with a probabilistic interpretation of the commonly-used kernel ridge regularization. Given independent and identically distributed inputs $x_i \in \mathbb{R}^d$ and outputs $y_i \in \mathbb{R}$ for $i = 1 \ldots n$, a Gaussian likelihood

$$y_i \mid (f, x_i) \sim \mathcal{N}(f(x_i), \tfrac{1}{2}\gamma), \quad p(y_i \mid f, x_i) \propto \exp(-\tfrac{1}{\gamma}(y_i - f(x_i))^2), \quad i = 1, \ldots, n,$$

is imposed, where $\gamma > 0$ is a hyperparameter representing the assumed noise level of the observations. In the noise-free setting, one can take $\gamma \to 0^+$. For notational convenience, let $X = (x_{ij})_{i,j=1}^{n,d} \in \mathbb{R}^{n \times d}$, and $Y = (y_i)_{i=1}^n \in \mathbb{R}^n$. To apply Bayes' theorem, the practitioner chooses a GP prior $f \sim \mathcal{GP}(m, \lambda^{-1}k)$, for $\lambda > 0$ a regularization hyperparameter. In the absence of prior information, it is common to choose $m \equiv 0$. To perform inference, the prediction and uncertainty for the output of a new input $x'$ is measured using the posterior predictive distribution (Rasmussen & Williams, 2006, Equation 2.19)

$$f(x') \mid (x_1, y_1), \ldots, (x_n, y_n) \sim \mathcal{N}(\bar{f}(x'), \lambda^{-1}\sigma(x')),$$

where $\bar{f}(x) = k_{x'}^\top(K_X + \lambda\gamma I)^{-1}Y$ and $\sigma(x') = k(x', x') - k_{x'}^\top(K_X + \lambda\gamma I)^{-1}k_{x'}$ for $k_{x'} = (k(x_i, x'))_{i=1}^n$ and $K_X = (k(x_i, x_j))_{i,j=1}^n$. In addition to the point estimate $\bar{f}(x)$, the posterior variance $k_{x'}$ can be taken as a calibrated measure of predictive uncertainty.

Kernel ridge regression (KRR) is a prediction method (with limited UQ) that estimates the output $f(x)$ for a given input $x$ by minimizing a regularized loss over a reproducing kernel Hilbert space (RKHS). Recall that every positive-definite kernel $k$ induces a RKHS, $H$, generated by the span of $\{k(x, \cdot)\}_{x \in \Omega}$ with norm $\|\cdot\|_H$ (see Appendix A for details). KRR considers estimators of the form:

$$\widehat{f} = \arg\min_{f \in H} \frac{1}{\gamma} \sum_{i=1}^n (f(x_i) - y_i)^2 + \lambda\|f\|_H^2. \tag{4}$$

From Kanagawa et al. (2018, Theorem 3.4), it turns out that $\widehat{f} = \bar{f}$, the mean predictor from the GP formulation, and so GP regression extends KRR to include an estimate of the uncertainty. Simultaneously, any optimization problem of the form (4) has a natural interpretation in terms of an underlying GP, providing uncertainty estimates for solutions to (4). We will make significant use of this relationship to construct our uncertainty-aware diagnostics.

## 2.3 Uncertainty and Diagnostics in GP

One of the advantages with treating a prediction task through the lens of statistical models is the capacity to analyse and estimate uncertainty. Bayesian methods, including GPs, naturally account for prediction uncertainty in the *posterior predictive distribution*, offering credible intervals for any quantile of uncertainty (Gelman et al., 1995). For GPs, the uncertainty about the prediction is contained in the posterior covariance kernel $x \mapsto \Sigma(x, x)$. Estimates of uncertainty are only as effective as the underlying model. Fortunately, the treatment of uncertainty often unlocks a wide array of diagnostic techniques, providing valuable feedback to the practitioner. In Bayesian statistics, the fundamental indicator of the quality of a particular prior is the *marginal likelihood*, also known as the *evidence*, given by

$$\mathcal{Z}_n = \mathbb{E}_{f \sim \mathcal{GP}(0, \lambda^{-1}k)}[p(Y \mid f, X)]. \tag{5}$$

This quantity can be thought of as the likelihood assigned by the prior to the observed data. It is typically convenient to work instead with the quantity

$$\mathcal{F}_n := -\log \mathcal{Z}_n = \frac{1}{2}Y^\top(K_X + \gamma I)^{-1}Y + \frac{1}{2}\log\det(K_x + \gamma I) - \frac{n}{2}\log\left(\frac{\lambda}{2\pi}\right), \tag{6}$$

which is called the (negative) *log-marginal likelihood*, alternatively the *Bayes free energy*. In practice, it is common to perform model selection by maximizing the marginal likelihood (equivalently, minimizing $\mathcal{F}_n$) with respect to hyperparameters of the prior. This is called an *empirical Bayes procedure* (Krivoruchko & Gribov, 2019), and it is the main inspiration of the PILE score which we introduce in Section 3. Aside from model selection, the free energy is also effective for model tuning. This process is referred to as *empirical Bayes* (Efron, 2024); and, provided that not too many parameters are tuned this way, it is often effective (Lotfi et al., 2022). For GPs, bandwidth tuning in the kernel, and the selection of the noise level $\gamma$, are both often conducted by minimizing Bayes free energy (Rasmussen & Williams, 2006, 5.4.1); see also Gribov & Krivoruchko (2020). However, information criteria—especially Bayesian ones—are not equivalent to test or cross-validation error, which leads to the question: when do they behave similarly? Fortunately, for GPs, it is known that the Bayes free energy behaves similarly to test error—at least when the number of training and test points are large; see, for example, Hodgkinson et al. (2023a); Luxburg & Bousquet (2004); Jin et al. (2022).

## 3 Physics-Informed Kernel Learning

The GP formulation of PIKL is based on a finite sample approximation of equation 2, using (possibly noisy) observations of the graph $(f, \mathcal{A}f)$ at points $x \in \Omega$. Following Doumèche et al. (2024a), we consider minimizers of the physics-informed empirical risk over $f$ in a RKHS $H$:

$$L_n(f) := \underbrace{\frac{1}{\gamma} \cdot \frac{1}{n} \sum_{i=1}^n (f(x_i) - y_i)^2}_{\text{data loss}} + \underbrace{\frac{1}{\rho}\|\mathcal{A}f\|_{L^2(\bar{\Omega}, \mu, \mathbb{R}^{p+1})}^2}_{\text{physics loss}} + \underbrace{\frac{1}{\eta}\|f\|_H^2}_{\text{regularization}}. \tag{7}$$

The first term is the *data loss*, prescribing adherence of $f$ to collected observations in the form of input-output pairs $(x_i, y_i)$, with $x_i \in \bar{\Omega}$ and $y_i \in \mathbb{R}$. These can be used in addition to, or in place of boundary conditions. The second term is the *physics loss*, which enforces that $f$ obey the prescribed equation $\mathcal{A}f = 0$. The third term biases the estimator towards a more regular solution, avoiding spikes and other singular behavior. For example, the RKHS $H_\nu$ associated with the Matern kernel of smoothness parameter $\nu > 0$ is equivalent to the Sobolev space $W^{\nu+d/2,2}(\Omega)$ (Wendland, 2004, Corollary 10.13), and so we have $\|f\|^2_{H_\nu(\bar{\Omega})} \asymp \|f\|^2_{W^{\nu+d/2,2}(\bar{\Omega})}$, the sum of the $L^2$ norm of the first $s$ derivatives of $f$. The temperatures $\gamma, \rho, \eta > 0$ are arbitrary and control the relative importance of these three terms. Since scaling $L^{\eta,\rho,\gamma}$ does not change its minimizer, it is typical to fix one parameter and vary the other two. For quantifying the uncertainty in model predictions pointwise in space, however, selecting the correct scale of $L^{\eta,\rho,\gamma}$ is required to have accurate and calibrated estimates of the pointwise posterior variance, and thus all three parameters are needed.

One approach to solving (7) using KRR, taken in Doumèche et al. (2024a;b), is to identify a new RKHS $H'$ with norm $\|f\|^2_{H'} = \|f\|^2_H + \frac{\eta}{2\rho}\|\mathcal{A}f\|^2_{L^2(\bar{\Omega}, \mu, \mathbb{R}^{p+1})}$. Alternatively, our approach relies on the observation that for a GP $f \sim \mathcal{GP}(m, k)$ over $\Omega$ supported on a Banach space $B \ni f : \Omega \to \mathbb{R}$, the pushforward of the process by a bounded linear operator $\mathcal{A} : B \to B'$ is itself a GP, supported on $B'$, with parameters

$$\mathcal{A}f \sim \mathcal{GP}(\mathcal{A}m, (\mathcal{A} \otimes \mathcal{A})k).$$

This observation has been applied numerous times in the literature to enforce PDE or other linear constraints on a GP via conditioning on the value of $\mathcal{A}f$ (Härkönen et al., 2023, Lemma 2.1), (Pförtner et al., 2024, Corollary 2), Macêdo & Castro (2010), Solin et al. (2018). While optimal in theory, the corresponding kernel of this space requires deep knowledge of the operator $\mathcal{A}$ and its eigenspectrum. Furthermore, samples drawn from a GP can be drastically less regular than the functions contained in the RKHS associated with its covariance kernel.[2]

To avoid this technical issue and to weaken our required assumptions on $k$, one can approximate equation 2 by a physics-informed version of KRR by estimating the $L^2$ norm using a quadrature rule $\{(w_i, z_i) : i = 1, \ldots, m\}$ for $\mu$ on $\bar{\Omega}$. One option is a Monte Carlo rule that selects $z_i$ uniformly at random over $\Omega$ and each $\Gamma_i$ with equal weighting $w_i = m^{-1}$. For improved precision, we opt for Gaussian quadrature rules. Our estimated loss function becomes

$$L_{m,n}(f) := \frac{1}{\gamma n}\sum_{i=1}^{n}(f(x_i) - y_i)^2 + \frac{1}{\rho}\sum_{i=1}^{m}w_i(\mathcal{A}f(z_i))^2 + \frac{1}{\eta}\|f\|^2_H. \tag{8}$$

The advantage of this approach is that, provided we can formulate $\{(f, \mathcal{A}f) : f \in H\}$ as a RKHS, (8) can be solved using the representer theorem. Let $H$ be a fixed RKHS with reproducing kernel $k$. For a multi-index $\alpha \in \mathbb{Z}_+^d$, we denote by $\partial_1^\alpha k(x, x')$ and $\partial_2^\alpha k(x, x')$ the iterated partial derivative of the first argument and second arguments, respectively: $\partial_1^\alpha \partial_2^\beta k(x, x') = \partial_{x_1}^{\alpha_1} \cdots \partial_{x_d}^{\alpha_d} \partial_{x'_1}^{\beta_1} \cdots \partial_{x'_d}^{\beta_d} k(x, x')$, where $\partial_{x_i}^{\alpha_i}$ denotes the $\alpha_i$-th partial derivative in $x_i$. To proceed, we require assumptions on $k$ and $\mathcal{A}$.

**Assumption 3.1** (Kernel Differentiability). Assume that $k$ has continuous $s$-th partial derivative, that is, for any multi-index $\alpha \in \mathbb{Z}_{\geq 0}^d$, $\|\alpha\|_1 \leq s$, $\partial_x^\alpha \partial_{x'}^\alpha k(x, x') \in C^0(\Omega \times \Omega)$.

**Assumption 3.2** (Bounded Coefficients). Assume that the coefficients $\{c_\alpha : \alpha \in \mathbb{Z}_{\geq 0}^d\}$ for $\mathcal{D}$ and each $\mathcal{B}_i$ are all uniformly bounded $\max_{\|\alpha\|_1 \leq s}\|c_\alpha\|_{L^\infty(\bar{\Omega})} \leq C < \infty$ for some $C > 0$.

By Proposition A.1 in Appendix A, under these assumptions, the RKHS associated with $k$ is contained in the image of $\mathcal{A}$. Indeed, as in Proposition A.3, the graph $\{(f, \mathcal{A}f) : f \in H\}$ itself an RKHS. Using this fact, in Theorem A.4, we obtain a representer theorem which can be used to solve (8). Theorem A.4 is the basis for our variant of the PIKL procedure. Given input-output pairs $\{(x_i, y_i)\}_{i=1}^{n}$ and a quadrature rule $\{(w_j, z_j)\}_{j=1}^{m}$, a solution to (8), and therefore an approximate solution to (7), can be obtained by finding the coefficients $(\alpha, \beta)$ that minimize (13). This procedure is general and particularly effective when $n$ and $m$ are not too large (Doumèche et al., 2024a).

---

[2]This can be seen by the fact that if the Cameron-Martin space associated with $\mathcal{GP}(m, k)$ is infinite dimensional, then it has zero Gaussian measure (Bogachev, 1998, Theorem 3.5.1). In other words, with probability one, $f \sim \mathcal{GP}(m, k)$ has $\|f\|_H = \infty$ whenever $H$ is not finite dimensional.

# 4 PHYSICS-INFORMED LOG EVIDENCE (PILE)

Now that we have demonstrated how PIML can be posed in terms of a KRR problem, we consider the interpretation of this finding in terms of GP regression, which encodes a natural notion of uncertainty. As discussed in Section 2.2, the estimator $\widehat{f}$ of Theorem A.4 coincides with the posterior predictive mean of a GP model. It can be verified that this model is prescribed by

$$(f, g) \sim \mathcal{GP}(0, \eta k_{\mathcal{A}}), \qquad y_i \mid f(x_i) \sim \mathcal{N}(f(x_i), \tfrac{1}{2}\gamma), \qquad r_j \mid g(z_j) \sim \mathcal{N}(g(z_j), \tfrac{1}{2}\rho w_j), \quad (9)$$

where each $r_j$ is interpreted as an observation of a boundary condition, which can be taken to be zero to enforce the constraint $\mathcal{A}f = 0$. Our formulation targets situations where a practitioner may also have potentially noisy access to the boundary data and forcing function in equation 1. In this case, one might take $r_j$ to be nonzero. Our goal is to remain as flexible as possible with respect to modes of data acquisition (i.e., possibly corrupted boundary data, interior observations of $f(x)$ or $\mathcal{D}f(x)$ at arbitrary $x \in \bar{\Omega}$) as well as with respect to the model used to represent $f$. This is a key benefit of the PIML approach.

The uncertainty in the prediction $\widehat{f}$ is encoded in the covariance of this GP: for $x', z' \in \bar{\Omega}$,

$$\mathrm{Cov}(f(x'), \mathcal{A}f(z')) = \eta \left( k_{\mathcal{A}}((x', z'), (x', z')) - \varsigma_{x',z'}^{\top} \Sigma_{m,n}^{-1} \varsigma_{x',z'} \right)$$

where

$$\Sigma_{m,n} = \begin{bmatrix} K_{xx} + \eta\gamma I_n & H_{xz} \\ H_{xz}^{\top} & G_{zz} + \eta\rho W^{-1} \end{bmatrix}, \quad \varsigma_{x',z'} = \begin{bmatrix} k(x', X) & (\mathsf{Id} \otimes \mathcal{A})k(x', Z) \\ (\mathsf{Id} \otimes \mathcal{A})k(X, z')^{\top} & (\mathcal{A} \otimes \mathcal{A})k(z', Z) \end{bmatrix}.$$

One of our main contributions of this work is to propose the negative log-marginal likelihood of this GP as an intrinsic uncertainty-aware measurement of model quality. We refer to this as the *Physics-Informed Log Evidence* (PILE). Inspired by standard protocol for tuning GPs, the PILE score doubles as a model selection criterion for optimizing hyperparameters, including $\rho$, $\gamma$, $\eta$, and any other hyperparameters for the kernel $k$, including bandwidth.

**Definition 4.1** (Physics-Informed Log Evidence (PILE)). The *Physics-Informed Log Evidence (PILE) criterion* is $\frac{2}{m+n}\mathcal{F}_{m,n}$ where $\mathcal{F}_{m,n}$ is the Bayes free energy of the GP equation 9:

$$\mathfrak{P}_{m,n} \coloneqq \frac{1}{m+n}\tilde{Y}^{\top}\Sigma_{m,n}^{-1}\tilde{Y} + \frac{1}{m+n}\log\det\Sigma_{m,n} + \log(2\pi\eta).$$

where $\tilde{Y} = (y_1, \ldots, y_n, r_1, \ldots, r_m)^{\top}$. The PILE criterion is to be interpreted as *lower is better*.

Unlike the empirical risk (8) which can be computed in quadratic time by the equation (13), computing the PILE score generally requires cubic time. Fortunately, for large $m, n$, several algorithms exist to compute the marginal likelihood quickly and under memory constraints Gardner et al. (2018); Ameli et al. (2025).

**Connection to the Fredholm Determinant.** Let us now consider the scenario where no data $(x_i, y_i)$ is prescribed and the PIKL framework is applied to find *a* solution $f \in H$ to $\mathcal{A}f = 0$. In this case, $r_i = 0$ can also be chosen and the PILE score simplifies to its last two terms. By taking $m \to \infty$, minimizers of (8) should become minimizers of our original problem (7). This is imposed by the following assumption.

**Assumption 4.2.** For any $f \in C^0(\bar{\Omega})$, as $m \to \infty$, the quadrature rule converges, that is, $\sum_{j=1}^{m} w_j f(z_j) \overset{m\to\infty}{\longrightarrow} \int_{\bar{\Omega}} f(z)\mathrm{d}\mu(z)$.

Our next objective is to show that as $m \to \infty$, the PILE score in this scenario converges in a suitable sense to a *Fredholm determinant*, which provides a surprising quantifier of the effectiveness of a particular choice of kernel $k$ to solve a given problem. The Fredholm determinant is a fascinating object with a complex history; and it typically appears only in the study of random matrices and determinantal point processes (Derezinski & Mahoney, 2021). We refer to Appendix B for its definition (and for the proof of the following result) and to Bornemann (2010) for details of its computation in practice.

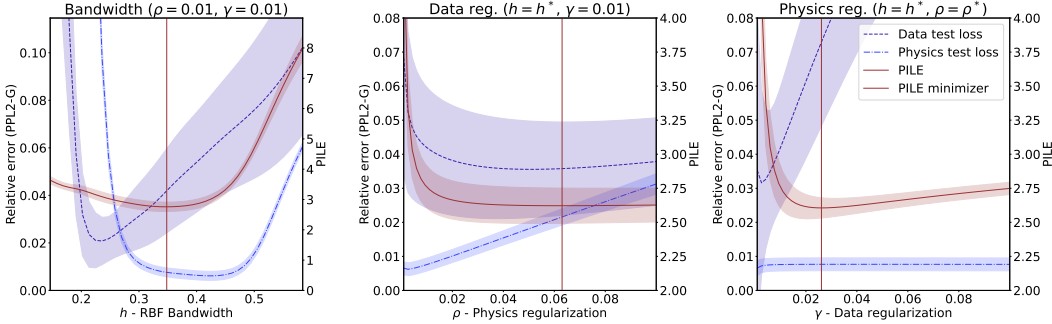

Figure 1: **Automatic hyperparameter selection with PILE.** PILE score and relative PPL2-G error sources for varying bandwidth, physics regularization, and data regularization parameters. Error bars show $\pm 2\widehat{\sigma}$ coverage, where $\widehat{\sigma}$ is the empirical standard deviation of PPL2-G (blue bars) and PILE (red bars). **(Left)** Bandwidth selection via minimizing the PILE score provides an accurate fit, balancing the data and physics generalization errors. **(Middle, Right)** After selecting the optimal bandwidth $h^*$, we sequentially minimize PILE first with respect to the physics regularization parameter $\rho$, then with respect to the data regularization parameter $\gamma$. For small values of $\rho$ and $\gamma$, PILE diverges as the regression model overfits the noisy observations.

**Theorem 4.3.** *Let* $\mathcal{G} : L^2(\bar{\Omega}, \mu, \mathbb{R}^{p+1}) \to L^2(\bar{\Omega}, \mu, \mathbb{R}^{p+1})$ *be the integral operator*

$$(\mathcal{G}f)(x, z) = \frac{1}{\eta\rho} \int_{\Omega \times \mathbf{\Gamma}} (\mathcal{A} \otimes \mathcal{A}) k(z, z') f(z') \mathrm{d}z'.$$

*Letting* $C_m = m\eta\rho - \sum_{i=1}^{m} \log w_i + m \log(2\pi\eta)$, *as* $m \to \infty$, *the sequence of normalized PILE scores converge to the* ***Fredholm determinant***: $m\mathfrak{P}_{m,0} - C_m \overset{m \to \infty}{\longrightarrow} \mathfrak{P}_0 = \log \det(I + \mathcal{G})$.

Note that one can also choose $\rho = \frac{1}{m\eta} \sum_{i=1}^{m} \log(\frac{w_i}{2\pi\eta})$ so that the normalized PILE score becomes equal to the PILE score. This choice of $\rho$ provides an avenue to calibrate uncertainty. Theorem 4.3 provides a new interpretation of the Fredholm determinant of integro-differential operators of the form of $\mathcal{G}$ in terms of the base model difficulty of solving differential equations within the corresponding RKHS induced by $k$. In turn, model selection for a given problem prescribed by the operator $\mathcal{A}$ can be achieved in a data-independent fashion by comparing values of $\mathfrak{P}_0$. In Appendix 5.2, we provide a case study, in which we use the Fredholm determinant to select the best kernel from a family of *anisotropic RBF kernels*. When used to solve a 1D convection PDE, we observe a drastic improvement in the capacity of the PIML model to fit the target solution.

## 5 CASE STUDIES

In this section we demonstrate how the PILE score can be used to select hyperparameters such as kernel bandwidth, loss regularization, and can even be used to select the kernel function from a parametrized family.

### 5.1 HYPERPARAMETER SELECTION USING THE PILE SCORE

Our first case study, shown in Figure 1, examines the baseline efficacy of the PILE score, and the Bayes free energy at large, for solving the multi-objective ambiguity in the PIML problem. The point of the study is to demonstrate the practicality of our method by using it to automatically select all the relevant hyperparameters for the problem. As a simple test case, we focus on solving a Poisson equation with Dirichlet boundary conditions, on the domain $\Omega = (-1, 1)^2$,

$$\Delta f(x) = g(x) \text{ for } x \in \Omega, \qquad f(x) = 0 \text{ for } x \in \partial\Omega, \qquad (10)$$

and with forcing function $g(x) = 10 + 10\sin(2\pi x)\sin(2\pi y)$. A 2D (type-1) Chebyshev quadrature scheme is used to determine the gridpoints $z_{(i,j)}$ and corresponding weights $w_{(i,j)}$ for $i, j = 1, \dots, m$ at which to evaluate the derivative loss: letting $s_i = \cos(\frac{2k+1}{2m_{\text{grid}}}\pi)$, we set $z_{(i,j)} = (s_i, s_j) \in \mathbb{R}^2$

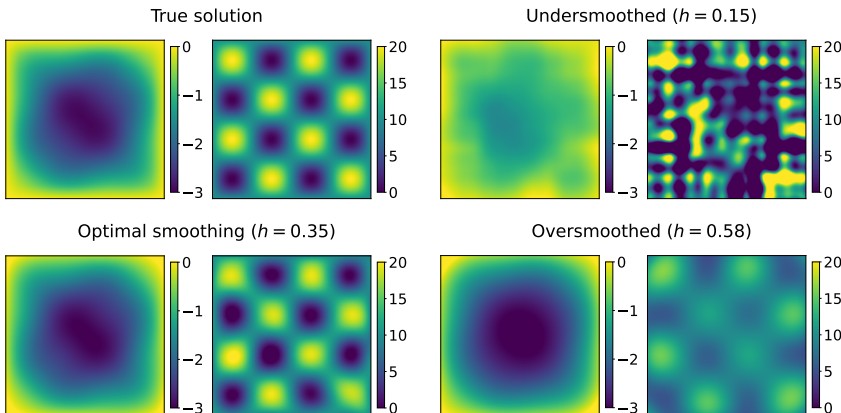

Figure 2: **Optimizing PILE prevents under- and over-smoothing.** Qualitative plot of the negative effects of oversmoothing and undersmoothing in PIKL. Each panel shows $\widehat{f}, \widehat{g}$ on the left and right, respectively. When $h$ is too small, the derivative estimate is undersmoothed and irregular. When $h$ is too large, oversmoothing effects prevent the model from fitting the derivative.

and $w_{(i,j)} = 4/m^2$. The type-1 Chebyshev grid points are known to have excellent properties for numerical approximation of integrals on $(-1, 1)$: for example, if $f, g \in C^\infty(\Omega)$, then the integration error $\|g - \mathcal{D}f\|_{L^2(\Omega)}$ converges as $e^{-\Omega(m)}$ (Trefethen, 2019, Chapters 7, 8). This choice of quadrature scheme enables us to closely approximate the true $L^2(\bar{\Omega})$ norm of the PDE residual.

In the physics informed setting, there are two sources of error: *data error*, measuring the fit of $\widehat{f}$ to $f$; and *physics error*, measuring the fit of $\widehat{g}$ to $g = \mathcal{D}f$. Following Hodgkinson et al. (2023a), we define the (unnormalized) *data PPL2-G* error as

$$\tilde{\mathcal{R}}_{\text{data}}(\widehat{f}) := \int_{\bar{\Omega}} \mathbb{E}_{\widehat{f}(z)}[(\widehat{f}(z) - f(z))^2] \, dz \approx \sum_{i,j=1}^{m_{\text{eval}}} w_i \mathbb{E}[(\widehat{f}(z_{(i,j)}) - f(z_{(i,j)})^2],$$

over $\widehat{f}(z) \sim \mathcal{N}(m^{|y,r}, \Sigma^{|y,r})$, marginalized over $\widehat{g}$. Analogously,

$$\tilde{\mathcal{R}}_{\text{phys}}(\widehat{g}) := \int_{\bar{\Omega}} \mathbb{E}_{\widehat{g}(z)} \left[ (\widehat{g}(z) - \mathcal{D}f(z))^2 \right] \, dx \approx \sum_{i,j=1}^{m_{\text{eval}}} w_i \mathbb{E}[(\widehat{f}(z_{(i,j)}) - f(z_{(i,j)})^2],$$

is the (unnormalized) *physics PPL2-G* error, marginalized over $\widehat{f}$. The quadrature points $z_{(i,j)}$ depend on $m_{\text{eval}}$, chosen to be large ($m_{\text{eval}} = 30 \gg m$) to ensure accurate $L^2(\bar{\Omega})$ approximations. For appropriate comparison, we normalize errors: $\mathcal{R}_{\text{data}}(\widehat{f}) = \frac{\tilde{\mathcal{R}}_{\text{data}}(\widehat{f})}{\|f\|_{L^2(\bar{\Omega})}}$, and $\mathcal{R}_{\text{phys}}(\widehat{f}) = \frac{\tilde{\mathcal{R}}_{\text{phys}}(\widehat{f})}{\|\mathcal{D}f\|_{L^2(\bar{\Omega})}}$. Both should be small, but since they cannot generally vanish simultaneously, *the optimal point on the Pareto front of these losses (the best solution to the PDE) is unclear.*

A typical challenge in RKHS and GP regression is selecting the *bandwidth* of a shift invariant kernel. A shift invariant kernel with bandwidth $h > 0$ has the form $k_h(x, y) = k(\frac{x-y}{h})$. These kernels enjoy special analytical properties and fast randomized approximations (Rahimi & Recht, 2007), making them popular in practice. It is important to tune the kernel bandwidth for optimal performance: if too large, the kernel becomes *oversmoothed* and the resulting regression estimator has high bias; whereas if it is too small, the resulting regression estimator may overfit observations or take near-zero values on unseen data. In this experiment, we train the regression model using grid size $n = m = 13$, so that there are a total of $13^2$ noisy observations $y_{(i,j)} = f(z_{(i,j)}) + \epsilon_{(i,j)}, z_{(i,j)} = \mathcal{D}f(z_{(i,j)}) + \epsilon'_{(i,j)}$ for independent $\epsilon_{(i,j)}, \epsilon'_{(i,j)} \sim \mathcal{N}(0, 1)$, with $i, j = 1 \dots 13$. Despite the high noise level and the relatively small number of samples, the regularized regression estimator matches the target function when fit with the optimal bandwidth according to PILE score; the PILE score has successfully overcome the multi-objective problem. When used to select $\rho, \gamma > 0$, divergence of the PILE score is an accurate indicator of model overfitting.

## 5.2 DIAGNOSING AND AVOIDING MODEL FAILURE WITH THE DATA-FREE PILE SCORE

In this case study, we analyze the well-known wave equation baseline introduced by Krishnapriyan et al. (2021), which was shown to break vanilla multilayer perceptron methods. We observe that for an

isotropic RBF kernel, there is no bandwidth that simultaneously achieves good physics and data fits, and PILE diagnoses this by selecting an oversmoothed 'all zero' solution. Perhaps more surprisingly, when we consider a broader class of 'anisotropic RBF' kernels, we find that optimizing the data-free PILE score (Figure 4) with respect to kernel function yields a model with excellent physics and data fit. The PILE-optimal kernel can be identified automatically, prior to data acquisition, and with no domain knowledge. Following (Krishnapriyan et al., 2021, Section 3.1), consider the convection PDE of the form

$$\begin{cases} \frac{\partial f}{\partial t}(t,x) + \beta \frac{\partial f}{\partial x}(t,x) = 0, & t \in [0,1], \ x \in [0, 2\pi], \\ f(0,x) = \sin(x). \end{cases} \tag{11}$$

We set $n = 1000$, $m = 20^2$, and assume access to observations of the form $y_i = f(x_i) + \epsilon_i$, $i = 1 \dots n$, for $x_i \sim \text{Unif}([0,1] \times [0, 2\pi])$. We observe empirically that fitting an RBF kernel as in Section 5 leads to pathological behavior, summarized in Figure 3. As shown in Figure 3 (left), the data loss is only small at bandwidths $h \approx 0.1$, while physics loss blows up at bandwidths $\leq h \approx 0.15$.

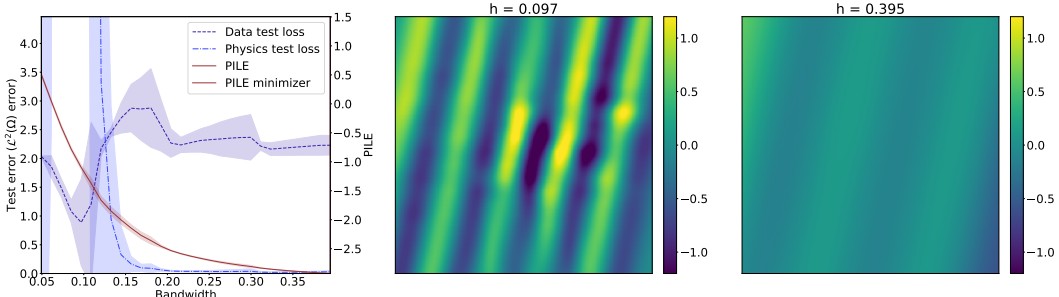

Figure 3: **PILE diagnoses model failure.** Fitting the convection PDE equation 11 with an RBF kernel. There is no appropriate bandwidth for this problem and the PILE score diagnosis this by selecting the 'all zeros' solution.

Instead, we could consider an *anisotropic* family of kernels defined by hyperparameters $\theta \in [-\pi, \pi]$, $s > 0$, and given by

$$k_{\theta,s}(x,y) := e^{-\frac{1}{2}(x-y)^T \Sigma_{\theta,s}(x-y)}, \quad \Sigma_{\theta,s} := \begin{bmatrix} \cos(\theta) & -\sin(\theta) \\ \sin(\theta) & \cos(\theta) \end{bmatrix} \begin{bmatrix} s^2 & 0 \\ 0 & s^{-2} \end{bmatrix} \begin{bmatrix} \cos(\theta) & -\sin(\theta) \\ \sin(\theta) & \cos(\theta) \end{bmatrix}.$$

To find an appropriate kernel among this family, we select $\theta, s$ to minimize the data free PILE score, whose loss landscape is shown in Figure 4. After kernel selection, the loss basins of the physics and data loss become drastically better conditioned, leading to an excellent model fit shown in Figure 4.

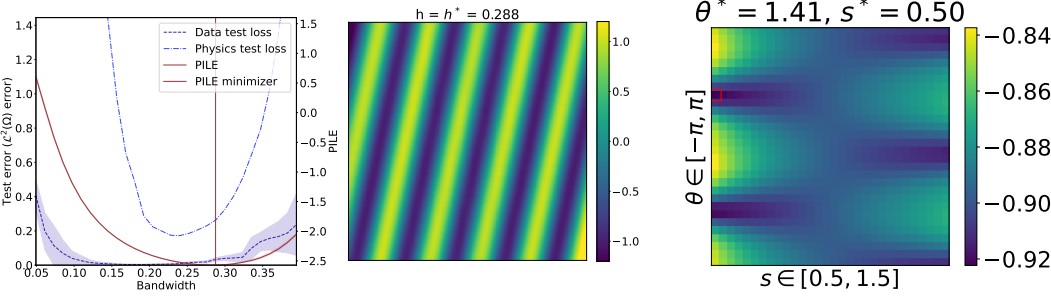

Figure 4: **Hyperparameter selection after kernel adjustment. (Left, Middle)** Fitting the convection PDE, equation 11, with an anisotropic RBF kernel. By choosing the kernel with minimum Fredholm determinant, we can automatically identify a "good" kernel for the continuity PDE. **(Right)** Fredholm determinants of $k_{\theta,s}$ plotted for $\theta \in [-\pi, \pi]$ and $s \in [0.5, 1.5]$. This quantity is empirically minimized at $\theta^* \approx 1.41$, $s^* \approx 0.5$ (shown in red).

## 6 CONCLUSIONS

We have introduced the PILE score, a model selection metric and diagnostic tool for physics-informed kernel learning to avoid failure modes. Our method amounts to hyperparameter selection via Empirical Bayes, thereby removing ambiguities involved in parameter selection under two competing loss terms. While already practical, it is useful to recognize how PILE can extend to other contexts.

- **Nonlinear Operators.** We studied linear differential operators with kernel-based models, but PIKL also extends to nonlinear PDEs (Chen et al., 2021), following the framework of Dashti et al. (2013). The nonlinear operator $\mathcal{A}$ is linearised via its Frèchet derivative $D\mathcal{A}$ at the solution $\widehat{f}$; and the PIKL setup and PILE score is applied using $D\mathcal{A}$ in place of $\mathcal{A}$. The justification for this PILE score follows the arguments of (Wacker, 2017).

- **Neural Networks.** The PIKL framework is distinct from both PINNs and Neural ODEs (since neural network regression differs from KRR). Approximations of the free energy are typically derived using Laplace approximations, such as in the derivation of the BIC (Schwarz, 1978), although extensions exist when these approximations fail (Drton & Plummer, 2017). Of particular relevance is the Interpolating Information Criterion (IIC) (Hodgkinson et al., 2023b), designed for models that interpolate data, akin to GP-based PIML. Inspired by IIC, one could evaluate the PILE score for trained PINNs using the empirical neural tangent kernel (Novak et al., 2022; Jacot et al., 2018) in place of $k$. At present, this is expensive, although approximations such as those seen in Ameli et al. (2025) suggests this might yet become tractable.

## 7 ACKNOWLEDGEMENTS

MWM would like to acknowledge the DOD DARPA AIQ program, the DOE LBNL LDRD program, and the NSF for partial support of this work. This material is based upon work supported by the U.S. Department of Energy, Office of Science, Office of Advanced Scientific Computing Research, Department of Energy Computational Science Graduate Fellowship under Award Number(s) DE-SC0023112.

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

## A    REPRODUCING KERNEL HILBERT SPACES

We provide a brief summary of reproducing kernel Hilbert spaces; for more details, we refer the interested reader to Steinwart & Christmann (2008). Let $H$ be a Hilbert space of functions $f : \mathcal{X} \to \mathbb{R}$ with an inner product $\langle \cdot, \cdot \rangle_H$. $H$ is called a *reproducing kernel Hilbert space* if for every $f \in H$, the evaluation functional $\iota_x : H \to \mathbb{R}$ defined for $x \in \mathcal{X}$ by $\iota_x f = f(x)$, is bounded, i.e., $|f(x)| \leq M_x \|f\|_H$ for some $M_x < +\infty$ and all $f \in H$. By the Riesz representation theorem, this implies that for any $x \in \mathcal{X}$, there exists $k_x \in H$ such that $f(x) = \langle f, k_x \rangle_H$. These elements $k_x$ are called the *feature maps*. The reproducing kernel of the Hilbert space is given by

$$k(x, y) = \langle k_x, k_y \rangle_H.$$

The Moore-Aronszajn theorem (Steinwart & Christmann, 2008, Theorem 4.21) implies that every positive-definite kernel induces a unique reproducing kernel Hilbert space, so the kernel completely defines $H$. One can also deduce the class of functions contained in the RKHS from the kernel. For example, if the kernel is differentiable, this implies that elements of $H$ are also differentiable, as in the following proposition.

**Proposition A.1** (Corollary 4.36 of Steinwart & Christmann (2008)). *Let $f \in H$. For $\alpha \le \lceil s \rceil$ the derivative $\partial^\alpha f(x) \in C^0(\Omega)$ exists and admits the bound:*

$$|\partial^\alpha f(x)| \le \|f\|_H \cdot (\partial_1^\alpha \partial_2^\alpha k(x,x))^{1/2}.$$

*Hence $\mathcal{D}f \in C^0(\Omega)$ for each $f \in H$.*

Furthermore, the image of $H$ under a linear operator is typically also an RKHS. The following proposition shows that the image of $H$ under a linear differential operator is an RKHS with a new kernel defined in terms of the operator.

**Proposition A.2.** *The space $G \coloneqq \{g \in H : \exists f \in H \text{ with } g(x) = \mathcal{D}f(x)\}$ endowed with the norm*

$$\|g\|_G = \inf_{f : \mathcal{D}f = g} \|f\|_H$$

*is the unique RKHS associated with the kernel*

$$g(x,y) \coloneqq (\mathcal{D} \otimes \mathcal{D})k(x,x').$$

*Moreover, $g(x,x') = \langle g_x, g_{x'} \rangle_H$ where the feature map $g_x \in H$ is given by $g_x(x') = (\mathcal{D} \otimes \mathsf{Id})k(x,x')$.*

*Proof.* It is sufficient to consider the case $\mathcal{D}f(x) = c_\alpha(x)\partial^\alpha f(x)$ for some $\alpha \in \mathbb{Z}$, as the general result follows from summing over terms of $\mathcal{D}f$. For any $\alpha$, it holds $y \mapsto \partial_x^\alpha k(x,y) \in H$ by repeated application of Lemma 4.34 of Steinwart & Christmann (2008). Hence,

$$\|(\mathcal{D} \otimes \mathsf{Id})(x,\cdot)\|_H = |c_\alpha(x)| \cdot \|(\partial^\alpha \otimes \mathsf{Id})k\|_H < \infty,$$

and

$$\begin{aligned}
\langle f, \mathcal{D}^* k_x \rangle_H &= c_\alpha(x)\langle f, (\partial^\alpha \otimes \mathsf{Id})k(x,\cdot) \rangle_H \\
&= c_\alpha(x)\partial_x^\alpha \langle f, k_x \rangle_H \\
&= \mathcal{D}f(x),
\end{aligned}$$

by the chain rule. $\qquad\square$

**Proposition A.3.** *The graph $\{(f, \mathcal{A}f) : f \in H\}$ is a RKHS $H_\mathcal{A}$ with norm*

$$\|(f, \mathcal{A}f)\|_{H_\mathcal{A}} = \|f\|_H$$

*and a multi-valued reproducing kernel $k_\mathcal{A}$ given by*

$$k_\mathcal{A}((x,z),(x',z')) = \begin{bmatrix} k(x,x') & (\mathsf{Id} \otimes \mathcal{A})k(x',z) \\ (\mathsf{Id} \otimes \mathcal{A})k(x,z') & (\mathcal{A} \otimes \mathcal{A})k(z,z') \end{bmatrix}.$$

Proposition A.2 is sufficient to prove Proposition A.3. To see this, note that the space $H_\mathcal{A} = \{(f, \mathcal{A}f) \in H \oplus H : f \in H\}$ can be equipped with the inner product acting only on the first coordinate: $\langle (f, \mathcal{A}f), (g, \mathcal{A}g) \rangle_{H_\mathcal{A}} = \langle f, g \rangle_H$. The evaluation functionals are guaranteed to be bounded by Proposition A.2, and the form of the kernel follows.

Here, we let $W = \text{diag}(w_i)_{i=1}^m$, so that $W^{1/2}$ can be interpreted as the elementwise square root.

**Theorem A.4** (Representer Theorem). *The minimizer $\widehat{f} \coloneqq \arg\inf_{f \in H} L_{m,n}(f)$ satisfies*

$$\widehat{f} \in \text{span}\left[ \{k(\cdot, x_i)\}_{i=1}^n \cup \{(\mathsf{Id} \otimes \mathcal{A})k(\cdot, z_j)\}_{j=1}^m \right], \tag{12}$$

*and for $\widehat{f} = \sum_{i=1}^n \alpha_i k(\cdot, x_i) + \sum_{j=1}^m \beta_j(\mathsf{Id} \otimes \mathcal{A})k(\cdot, z_j)$, the coefficients $(\alpha, \beta) \in \mathbb{R}^{m+n}$ minimize*

$$\frac{1}{\gamma n}\|Y - K_{xx}\alpha - H_{xz}\beta\|_2^2 + \frac{1}{\rho}\|W^{1/2}(H_{xz}^\top \alpha - G_{zz}\beta)\|_2^2 + \frac{1}{2\eta}\begin{bmatrix} \alpha \\ \beta \end{bmatrix}^\top \begin{bmatrix} K_{xx} & H_{xz} \\ H_{xz}^\top & G_{zz} \end{bmatrix}\begin{bmatrix} \alpha \\ \beta \end{bmatrix}, \tag{13}$$

*which is now equal to $L_{m,n}(\widehat{f})$, where $[K_{xx}]_{i,j} = k(x_i, x_j)$, $[H_{xz}]_{ij} = (\mathsf{Id} \otimes \mathcal{D})k(x_i, z_j)$, and $[G_{zz}]_{ij} = (\mathcal{D} \otimes \mathcal{D})k(z_i, z_j)$.*

*Proof of Theorem A.4.* Property equation 12 follows from the fact that the $\|\cdot\|_H$ projection of any $f \in H$ onto the span does not affect the values of $f(x_i)$ nor $\mathcal{D}f(z_j)$). By the representer property for $H$,

$$\langle k_{x_i}, g_{z_j}\rangle_H = \langle k_{x_i}(\cdot), (\mathsf{Id} \otimes \mathcal{D})k(\cdot, z_j)\rangle_H = (\mathsf{Id} \otimes \mathcal{D})k(x_i, z_j),$$

and by Proposition A.2 $\langle g_{z_i}, g_{z_j}\rangle = (\mathcal{D} \otimes \mathcal{D})k(z_i, z_j)$. Plugging these identities into $\|\widehat{f}\|_H^2$ yields

$$\|\widehat{f}\|_H^2 = \begin{bmatrix} \alpha \\ \beta \end{bmatrix}^\top \begin{bmatrix} K_{xx} & H_{xz} \\ H_{xz}^\top & G_{zz} \end{bmatrix} \begin{bmatrix} \alpha \\ \beta \end{bmatrix}.$$

Eqn equation 13 is Eqn. equation 12 rewritten in terms of $\alpha, \beta$. $\qquad\square$

## B  FREDHOLM DETERMINANTS

Let $H$ be a separable Hilbert space with an inner product $\langle\cdot,\cdot\rangle_H$ and let $\mathcal{K}(H)$ denote the space of compact linear operators $A : H \to H$. The spectrum

$$\mathrm{spec}(A) = \{\lambda \,:\, \ker(\lambda I - C) \neq \{0\}\},$$

of any $A \in \mathcal{K}(H)$ is countable and can accumulate only at zero. Hence, there is a (possibly infinite) sequence of eigenvalues $\{\lambda_n(A)\}_n$. For any $A \in \mathcal{K}(H)$, let

$$\sigma_1(A) \geq \sigma_2(A) \geq \cdots > 0,$$

denote the ordered singular values of $A$, defined as the square root of the eigenvalues of $A^*A$.

**Definition B.1.** A compact operator $A \in \mathcal{K}(H)$ is *trace class* if $\sum_n \sigma_n(A) < +\infty$.

By Holder's inequality, for any trace class operator $A$, the Schatten norm satisfying

$$\|A\|_p^p = \sum_n \sigma_n(A)^p, \qquad p \geq 1,$$

is finite. Using the definition to verify whether an operator is trace class is a near-impossible task in general. Fortunately, the following theorem provides conditions that guarantee in our kernel setting that all operators are trace class.

**Theorem B.2.** *Let $k$ be a symmetric positive-definite kernel on $\Omega \times \Omega$ such that $x \mapsto k(x, x) \in L^2(\Omega)$ for $\Omega \subseteq \mathbb{R}^d$. The integral operator $K : L^2(\Omega) \to L^2(\Omega)$ defined by*

$$(Kf)(x) = \int_\Omega k(x, y)f(y)\mathrm{d}y, \qquad \textit{is trace class.}$$

We are now ready to define the Fredholm determinant of an operator.

**Definition B.3.** The *Fredholm determinant* of a trace class operator $A$ is given for $\rho > 0$ by

$$\det(I + \rho A) = \prod_n (1 + \rho s_n(A)).$$

It is straightforward to verify that the Fredholm determinant can only be defined for a trace class operator, since (Knopp, 2013, p. 232)

$$1 + \rho\|A\|_1 \leq \det(I + \rho A) \leq \exp(\rho\|A\|_1).$$

Note that for $H = \mathbb{R}^m$, the Fredholm determinant reduces to the standard determinant, as a bounded linear operator $A : \mathbb{R}^m \to \mathbb{R}^m$ can be represented as a matrix $A_m$ over the basis elements $\{e_i\}_{i=1}^m$ and

$$\det(I + \rho A) = \prod_{n=1}^m (1 + \rho s_n(A)) = \det(I + \rho A_m).$$

In the case of integral operators, the Fredholm determinant is expressed in terms of the infinite series (Bornemann, 2010, eqn. (3.7))

$$\det(I + \rho K) = 1 + \sum_{n=1}^\infty \frac{\rho^n}{n!} \int_{\Omega^n} \det(k(x_i, x_j))_{i,j=1}^n \mathrm{d}x_1 \cdots \mathrm{d}x_n.$$

*Proof of Theorem 4.3.* Starting from the PILE score,

$$
\begin{aligned}
m\mathfrak{P}_{m,n} &= \log\det(G_{zz} + \eta\rho W^{-1}) + m\log(2\pi\eta) \\
&= \log\det(W^{1/2}G_{zz}W^{1/2} + \eta\rho I) + m\log(2\pi\eta) - \sum_{i=1}^{m}\log w_i \\
&= \log\det(I + W^{1/2}(\eta\rho)^{-1}G_{zz}W^{1/2}) + C_m,
\end{aligned}
$$

and consequently,

$$
m\mathfrak{P}_{m,n} - C_m = \log\det(I + W^{1/2}((\mathcal{A}\otimes\mathcal{A})k(z_i,z_j))_{i,j=1}^{m}W^{1/2}).
$$

The result now follows from (Bornemann, 2010, Theorem 6.1). □

