# OpenReview forum: "Uncertainty-Aware Diagnostics for Physics-Informed Machine Learning"
_ICLR.cc/2026/Conference — ICLR 2026 Poster_

### Official Review · Reviewer_hS2h · 2025-10-29

**Soundness:** 4
**Presentation:** 3
**Contribution:** 4
**Rating:** 6
**Confidence:** 4

**Summary:**

This paper introduces the Physics-Informed Log Evidence (PILE), a novel, uncertainty-aware model selection criterion for Physics-Informed Machine Learning (PIML). The work is situated within the Physics-Informed Kernel Learning (PIKL) framework, which uses Gaussian Processes (GPs) to solve linear PDEs. The key insight is to use the Bayesian marginal likelihood (or Bayes free energy) to navigate the multi-objective trade-off between data fidelity and physics constraints—a common pain point in PIML. The authors demonstrate that minimizing the PILE score effectively selects hyperparameters (e.g., kernel bandwidth, regularization weights) and even the kernel function itself. A particularly innovative contribution is the "data-free" PILE, which connects to Fredholm determinants and allows for a priori kernel selection suited to a given PDE, before any data is observed. Empirical results on Poisson and convection equations show that PILE reliably diagnoses and prevents common PIML failure modes.

**Strengths:**

This paper makes a clear and original contribution by introducing the Physics-Informed Log Evidence (PILE) score as an uncertainty-aware diagnostic and model selection principle for physics-informed machine learning. The work is significant in that it addresses the long-standing ambiguity in balancing data loss and physics loss in PIML, by providing a single, principled metric rooted in Gaussian process regression. A notable strength is the dual perspective: the authors present both a priori diagnostics (via a Fredholm determinant in the data-free case) and a posteriori diagnostics (via PILE after training).

The quality of the work is high. The theoretical development is rigorous, linking PIML to marginal likelihood in GPs, and the empirical case studies (Poisson, wave, and convection PDEs) clearly show how PILE can guide hyperparameter selection, prevent over/under-smoothing, and diagnose failure modes. The exposition is well-structured and accessible, with illustrative figures that support the main claims. Overall, the paper advances originality by integrating uncertainty quantification into model evaluation, and its significance lies in offering a practical tool for robust diagnostics in PIML.

**Weaknesses:**

The primary limitation is that the methodology is restricted to kernel-based Gaussian process models. While the authors outline possible extensions to neural network–based PIML methods (e.g., PINNs, neural operators), these are not demonstrated. Empirical validation on nonlinear PDEs or higher-dimensional benchmarks would substantially strengthen the generality claims.

A second limitation is computational scalability. The PILE score requires cubic time in the number of data and quadrature points, which the authors acknowledge. They cite approximate methods for marginal likelihood computation, but no experiments demonstrate that these approximations keep PILE practical at scale. More discussion or demonstrations here would increase the impact.

Finally, while the connection to the Fredholm determinant is mathematically elegant and provides a novel perspective on data-free kernel selection, its practical utility is demonstrated only in a single anisotropic kernel case study. Further evidence that this insight generalizes would help substantiate its significance.

**Questions:**

How does the computational cost of PILE scale with both the dimension of the PDE domain 𝑑 and the number of data/quadrature points What are the practical limits for its application in higher-dimensional or large-scale problems?

The extension to neural networks via the NTK is mentioned. Could you elaborate on the practical challenges of computing or approximating the PILE score in this setting, given the known limitations of the NTK approximation in finite-width networks?

How sensitive is the PILE score to the choice and quality of the quadrature scheme used to approximate the physics loss? Would different quadrature methods or levels of accuracy substantially change the diagnostic outcome?

In the data-free kernel selection, the Fredholm determinant is used. Are there intuitive interpretations of its value that could help a practitioner understand why one kernel is “better adapted” to a PDE than another?

Is code available to reproduce the experiments and facilitate the adoption of the PILE score by the community?

---

> ### Author Response · Authors · 2025-12-03
> **Response to Reviewer hS2h**
>
> We thank the reviewer for their thorough reading and thoughtful questions.
>
> **On computational scaling**: In the naive setting, evaluating PILE involves cubic complexity in the number of data and quadrature points, similar to standard GP inference. While approximate marginal likelihood methods exist, in our experiments they were not required: because Gaussian quadrature converges rapidly in low spatial dimensions, only a modest number of points was typically sufficient for stable PILE estimates. As $d$ increases, the intrinsic limitations of kernel-based solvers tend to arise sooner than any practical bottleneck induced specifically by PILE.
>
> **On the NN extension and NTK considerations**: The primary practical difficulty lies in generating the NTK itself, whose cost scales with the number of parameters. The limitations of NTK-based approximations mostly affect model dynamics during training. At the end of training, the local approximation arising from the NTK approximation seems to be appropriate as to estimate a Bayesian posterior. For more details here, we refer to [1]. We did not include a neural network demonstration because doing so would require substantial additional exposition and careful case-study design. Moreover, the impact of architectural choices in PINNs is far less regular than for kernel methods, making it harder to isolate the benefits of PILE in a concise example.
>
> **On the sensitivity to quadrature scheme**: PILE is sensitive to quadrature only to the degree that the quadrature approximates the relevant operator integrals. Using Gaussian quadrature, we found that the score stabilizes rapidly as the number of points increases. Adding excessively many points can make the underlying kernel problem harder to solve numerically, but we did not observe changes in ranking or diagnostics once moderate resolution was reached.
>
> **On intuitive interpretations of the Fredholm determinant**: This is quite challenging, and hence the reason for our careful discussion of the topic, but we see the Fredholm determinant in this context as reflecting the regularity of the problem at hand. In our convection example (the second case study), the solution exhibits different characteristic scales along each coordinate. An isotropic kernel mismatches these scales, whereas an anisotropic kernel aligns with them. The determinant captures this alignment, as kernels that encode the correct structural bias yield larger capacity to represent solutions, leading to a more favourable determinant value. This interpretation is what the case study was designed to illustrate.
>
> **Regarding code availability**: Code has been provided in the attached Supplementary Material. We also have a GitHub repository for this project that can be shared in the camera-ready version.
>
> We thank the reviewer again for their constructive assessment.
>
> [1] Wilson, J., van der Heide, C., Hodgkinson, L., & Roosta, F. (2025). Uncertainty quantification with the empirical neural tangent kernel. arXiv preprint arXiv:2502.02870.

---

### Official Review · Reviewer_Pt1Q · 2025-11-01

**Soundness:** 3
**Presentation:** 2
**Contribution:** 2
**Rating:** 4
**Confidence:** 3

**Summary:**

This paper presents a metric, named PILE, to evaluation principle for hyperparameters for physics informed models. The metric is for kernel learning only.

**Strengths:**

1. The paper addresses an important topic for incorporating the knowledge into data-driven learning.
2. Thorough theoretical treatment of the problem.

**Weaknesses:**

1. Applicable to kernel learning only. It's still an open problem whether it can be extended to other ML techniques especially neural networks. Although kernel learning is highly capable, there are still a large selection of NN-based ML methods which would greatly benefit from physics knowledge.
2. The organization and exposition of the paper can be further improved. It is good to be mathematically rigorous, the exposition can be improved to give the readers more intuition, instead of piles of math equations.

**Questions:**

1. I have concerns over the phrase "multi-objective nature of the approach" of balancing both the data loss and physical constraints. Although it is true that the two components may draw the overall loss to different ends of the loss function spectrum, the ultimate goal is to train a model most generalizable. In my view, the multi-objectiveness is only the superficial aspect of the optimization.
2. It appears that the metric is only applicable to physical systems modeled by PDEs, hence the background information in section 2.1. Would the method applicable to other systems, such as differential-algebraic equations?
3. Issue with Fig 1, although the general trend of three component is very clear, the magnitude as shown in the y-axis is very small. Would the numerical noise affect the solution?

---

> ### Author Response · Authors · 2025-12-03
> **Response to Reviewer Pt1Q**
>
> We thank the reviewer for their careful examination of our work. While we understand the trepidation here, we hope that the comments from the other reviewers, as well as some clarifications can help address these concerns.
>
> **On applicability beyond kernel learning**: While our current formulation relies on Gaussian processes to obtain a tractable evidence objective, we do outline in the conclusion how analogous ideas could be extended using other appropriate approximate marginal likelihoods. Experiments have not been included, as they are out-of-scope and require significantly more work and discussion to perform, but there are otherwise no fundamental limitations here.
>
> **On exposition and organization**: While we appreciate the concern, the theoretical development requires several non-standard components (physics-informed kernels, operator-based regularization, and marginal likelihood computation) which motivated the level of mathematical detail. This poses challenges within the conference format, but have attempted to reach an ideal balance. Other reviewers (e.g. Reviewer Jkyr) have been more positive in their impressions of the presentation, but did also acknowledge that several aspects may be abstract for general audiences that are unfamiliar with the kernel setting. We have made an attempt to summarise this material in the Appendix for newcomers as well.
>
> **On the “multi-objective” nature of PIML**: This is an important point. "Most generalizable" has no concrete definition, since it is unclear whether the data is more reliable than the physics knowledge; in many cases, it is not. This is different from other ML settings where the test data can be considered essentially truthful. The viewpoint that this multi-objective nature is superficial, and need only be treated from the regularisation point of view, has unfortunately given rise to many numerical problems in the literature; see [1] for example.
>
> **On applicability to other system types (e.g., DAEs)**: Yes, the method can be applied to other classes of problems. The differential-algebraic equation setting is a subset of the nonlinear PDE setting, which we have discussed in the conclusion.
>
> **On Fig. 1 and numerical noise**: The magnitudes shown are small because the system under study has low uncertainty scales once conditioned on data and physics. The plotted error bars explicitly reflect posterior uncertainty, so any numerical noise is already incorporated into the visualization.
>
> We appreciate the reviewer's engagement with the ideas and thank them again for their comments.
>
> [1] Krishnapriyan, A., Gholami, A., Zhe, S., Kirby, R., & Mahoney, M. W. (2021). Characterizing possible failure modes in physics-informed neural networks. Advances in neural information processing systems, 34, 26548-26560.

---

### Official Review · Reviewer_hf97 · 2025-11-02

**Soundness:** 3
**Presentation:** 3
**Contribution:** 3
**Rating:** 6
**Confidence:** 3

**Summary:**

The paper addresses the problem of measuring the model quality in the case of physics-informed machine learning (PIML). Since PIML companies data and physical constraints (like differential equations) to train models, this leads to multi-objective approach which is hard to evaluate. Current methods can produce models that look good by standard metrics but still fail in surprising ways due to poor understanding of epistemic uncertainty. The authors work within a Gaussian process regression framework and introduce the Physics-Informed Log Evidence (PILE) score. The PILE score provides a single, uncertainty-aware metric that bypasses the ambiguities of traditional test losses. The paper addresses a critical gap in PIML methodology.

**Strengths:**

- The multi-objective evaluation issue is an important problem in PIML that practitioners struggle with. Having a principled diagnostic is valuable.
- A single metric for hyperparameter selection is very usable (as opposed to juggling multiple competing objectives such as data loss vs. physics loss vs. test error).

**Weaknesses:**

- The current set-up is limited to Gaussian processes (not major, but it would obviously great to have something model agnostic).
- This also limits the problems in which PILE would be useful to rather smaller scale problems, as GPs don't scale well. The PILE score itself introduces additional complexity. Hence, it is not fully clear what is practical limitations in terms of the computational cost and up to what degree it would be feasible to use it in practice.

**Questions:**

- How would PILE score behave when a model completely violates a hard constrain and when it has an ok residual error in a soft constraint? Is is possible to distiguish between hard and soft constraints?

- How does the computational cost of evaluating the PILE score scale as constraint complexity increases? What are computational limitations? Or in other words, up to what dimensionality is it reasonable to use it (given it's higher computational complexity and cubic complexity of GPs)?

---

> ### Author Response · Authors · 2025-12-03
> **Response to Reviewer hf97**
>
> We thank the reviewer for taking the time to read our work and for their positive assessment.
>
> **On the use of Gaussian processes**: The restriction to Gaussian processes arises from the need for a tractable marginal likelihood, which is central to defining the PILE score. While this limits the current formulation, we discuss in the conclusion how analogous ideas extend to broader classes of physics-informed models.
>
> **On practicality and scalability**: At present, the work is generally constrained to cases where kernel learning is most appropriate. However, this covers a rather substantial number of use cases, and kernel learning is becoming increasingly used in PDEs as a mesh-free alternative. We believe that the utility of the PILE score is profound in this context. The additional complexity required by the PILE score in these contexts is minimal, and we have found the calculations straightforward in practice.
>
> **Distinguishing hard and soft constraints**: Indeed, this is the primary advantage of the PILE score! It is often the case that the hard constraint is softened, and it becomes ambiguous how to balance the two losses. We have found the PILE score does an excellent job of distinguishing the two, as seen in our first case study.
>
> **Computational cost and dimensionality**: The dominant computational factors are the usual GP cubic scaling in the number of training points and the number of quadrature points used to discretize the constraints. Thanks to the rapid convergence of Gaussian quadrature in low spatial dimensions, only a small number for $m$ is typically required. For this reason, we view the method as most suitable for problems up to a few spatial dimensions, which fortunately covers virtually all PDE-driven applications where kernel methods are employed.
>
> We thank the reviewer again for their constructive feedback and interest in our method.

---

### Official Review · Reviewer_Jkyr · 2025-11-03

**Soundness:** 4
**Presentation:** 4
**Contribution:** 4
**Rating:** 8
**Confidence:** 3

**Summary:**

This paper proposes PILE (Physics-Informed Log Evidence), a principled and uncertainty-aware model selection metric for Physics-Informed Machine Learning. Working within a Gaussian Process-based Physics-Informed Kernel Learning framework, the authors show that PILE resolves the core multi-objective tension between data-fit and physics-fit by providing a single score that quantifies model quality, enabling reliable tuning of kernel hyperparameters, regularization weights, and noise levels. They further show that in the absence of data, PILE converges to a Fredholm determinant, which can be used to select kernels a priori that are well-adapted to a given PDE. Through case studies, including a challenging convection PDE known to cause PIML failures, the paper demonstrates that PILE not only identifies when a model or kernel is misspecified, but also guides the choice of kernels and parameters that yield high-quality solutions, thereby offering a robust diagnostic and model-selection tool for PIML.

**Strengths:**

The paper offers a novel contribution to physics-informed machine learning by introducing PILE, a principled and uncertainty-aware model selection criterion based on Bayesian marginal likelihood. Unlike existing PIML approaches that rely on heuristic loss weighting or manual hyperparameter tuning, this work reframes the problem through Bayesian evidence maximization, providing a single score that jointly captures data-fit, physics-fit, model complexity, and uncertainty calibration. A particularly original aspect is the theoretical result showing that, in the absence of data, the PILE score converges to a Fredholm determinant of a PDE-informed integral operator. This reveals a connection between PIML, Gaussian processes, and spectral operator theory, opening a new line of investigation.

The paper is technically strong, with a rigorous derivation of PILE from the Bayes free energy of the physics-informed GP model. The theory is well supported by clear assumptions and proofs, and the empirical results, though modest in scale, effectively illustrate the practical value of the method. In particular, the convection PDE case study demonstrates that PILE can identify kernel choices that significantly improve solution quality, underscoring the method’s practical utility.

The presentation is generally clear and well organized. The progression from standard GP regression to physics-informed kernel learning and finally to the PILE formulation is logical and accessible for readers with a background in GPs or kernel methods. While some of the operator-theoretic aspects may remain abstract for a broader audience, the core ideas are well communicated and supported by helpful examples.


This work is significant because it directly tackles a key limitation of current PIML methods: the lack of a principled mechanism for model and hyperparameter selection. By enabling both data-informed and data-free kernel selection, the proposed framework has the potential to meaningfully reduce reliance on trial-and-error and improve robustness in scientific ML workflows. The core ideas are sufficiently general that they could influence future developments beyond GP-based methods, including uncertainty-aware neural operators and physics-based deep learning. Overall, the paper makes a meaningful, well-founded, and potentially impactful contribution to the field.

**Weaknesses:**

While the paper is strong overall, a few areas could be improved to enhance its practical impact. The work is primarily method-driven and introduces a novel and well-motivated framework, and I did not identify major weaknesses in the core methodology or theoretical development. My comments are therefore more about opportunities to further strengthen the empirical validation. The experiments are limited to relatively low-dimensional PDEs with simple settings e.g., the main results focus on a 1D Poisson equation and a 1D convection problem and demonstrating the method on more challenging benchmarks (e.g., 2D/3D heat equation, 2D convection-diffusion equation etc.) would better showcase its broader applicability. In addition, the connection to Fredholm determinants is theoretically elegant but only demonstrated through a single kernel-selection case study. Further analysis of stability and generality across PDE types and kernels would help strengthen confidence in its practical utility.

**Questions:**

1. In the data-free PILE formulation, the theory requires $m\rightarrow\infty$ for convergence to the Fredholm determinant. In practice, how do the authors recommend choosing $m$ for a given PDE and kernel? Are there heuristics for determining when $m$ is "large enough" for the PILE score to be reliable?

2. Have the authors observed how sensitive the PILE score is to the number and distribution of collocation/quad points? For example, do the kernel rankings stabilize beyond a certain $m$? Any such empirical guidance would be useful for practitioners.

---

> ### Author Response · Authors · 2025-12-03
> **Response to Reviewer Jkyr**
>
> We thank the reviewer for their very positive assessment and thoughtful comments! We are pleased that the motivation, theoretical development, and empirical choices were clearly understood and appreciated.
>
> **On experimental scope**: We agree that extending the study to higher-dimensional PDEs would be interesting. Our decision to focus on lower-dimensional settings was deliberate: these allow clear visual comparisons between solutions and PILE rankings, which we have found to be the most transparent way of illustrating the behaviour of the score. Higher-dimensional examples produce PILE values but do not readily support interpretable visual diagnostics within limited space, which is why we opted for settings where readers can directly assess solution quality. We hope this helps clarify our rationale for the experimental design.
>
> **On convergence of the data-free PILE and selection of $m$**: As shown in Bornemann [1], Fredholm determinant approximations under Gaussian quadrature converge very rapidly, and we observe the same effect empirically for PILE. In practice, only a small number of collocation points is typically needed. A simple and effective heuristic that is analogous to mesh refinement in numerical solvers is to increase $m$ until the PILE score stabilizes. In our experiments, kernel rankings generally stabilized after relatively few points, and the score was robust to variations in both the number and placement of quadrature points.
>
> We thank the reviewer again for their careful reading and constructive suggestions.
>
> [1] Bornemann, F. (2010). On the numerical evaluation of Fredholm determinants. Mathematics of Computation, 79(270), 871-915.

---

### Meta-Review · Area_Chair_86mp · 2025-12-30

**Summary:**

I find this paper makes a clear and worthwhile contribution to physics-informed machine learning by introducing PILE, a Bayesian marginal likelihood criterion for model selection in kernel-based PIML. The central insight—that the multi-objective tension between data fidelity and physics constraints can be resolved through a single evidence-based score—is both theoretically grounded and practically useful. The connection to Fredholm determinants for data-free kernel selection is mathematically elegant and provides a new perspective on a priori model assessment.

The experimental demonstrations are limited to low-dimensional PDEs, but convincingly show that PILE tracks test error and successfully guides hyperparameter selection. The convection PDE case study directly addresses known failure modes in PIML, showing both diagnostic capability (identifying when isotropic kernels fail) and prescriptive value (selecting anisotropic kernels via data-free PILE). The restriction to GPs is a genuine limitation that bounds the method's applicability, but within this scope the contribution is solid. I am persuaded that this work advances the state of GP and kernel-based physics-informed learning, even though extensions to neural networks remain speculative at best.

**Reviewer Concerns:**

The primary methodological concern—restriction to kernel/GP methods—was acknowledged by the authors and is inherent to the approach. Tractable marginal likelihood requires the GP framework, and while the authors outline potential extensions via neural tangent kernels, these are undemonstrated. I consider this an honest limitation rather than a flaw.

Concerns about experimental scope (low-dimensional PDEs only) were addressed by the authors' explanation that such settings enable interpretable visual verification of solution quality. I find this partially satisfying; it would strengthen the paper to show PILE values stabilize appropriately in higher dimensions, but the current demonstrations suffice to establish the method works.

Reviewer Pt1Q's concern that the multi-objective nature is "superficial" seems to reflect a misunderstanding of the problem. The paper explicitly addresses why regularization-based views are insufficient, citing the failure modes documented by Krishnapriyan et al. (2021). I discount this concern. The question about Figure 1 magnitudes ignores the error bars already shown, which incorporate posterior uncertainty.

The scalability question (computational cost at large n and m) remains partially unresolved. The authors note rapid quadrature convergence in low dimensions but do not demonstrate approximate marginal likelihood methods. For the kernel setting targeted by this work, where problem sizes are typically moderate, I find the practical utility is established even without large-scale demonstrations.

**Reviewer Scores:**

Reviewer Jkyr: 8 → 8. Strong technical review with no fundamental concerns. Experimental scope question adequately addressed.

Reviewer hf97: 6 → 6. Hard/soft constraint confusion clarified, but scalability concerns remain valid. No change expected.

Reviewer Pt1Q: 4 → 5. Some concerns addressed, but the fundamental misunderstanding about multi-objective optimization limits engagement. This review carries reduced weight given its quality issues.

Reviewer hS2h: 6 → 6-7. Substantive responses to all questions. May increase slightly given thorough engagement, but scalability and scope concerns persist.

---

### Decision · Program_Chairs · 2026-01-26

Accept (Poster)